# Host Immunity Mechanisms Against Bacterial and Viral Infections in *Bombyx mori*

**DOI:** 10.3390/insects16111167

**Published:** 2025-11-15

**Authors:** Sadaf Ayaz, Wei-Wei Kong, Jie Wang, Shi-Huo Liu, Jia-Ping Xu

**Affiliations:** 1Anhui Province Key Laboratory of Resource Insect Biology and Innovative Utilization, School of Life Sciences, Anhui Agricultural University, Hefei 230036, China; fazalsadaf1122@gmail.com (S.A.); m18505637859@163.com (W.-W.K.); 2Anhui International Joint Research and Developmental Center of Sericulture Resources Utilization, Hefei 230036, China; 3Institute of Sericulture, Anhui Academy of Agricultural Sciences, Hefei 230036, China; wangjie_3001@163.com

**Keywords:** silkworm, immunity, bacteria, virus, antimicrobial peptides (AMPs), pattern recognition receptors (PRRs), RNA interference (RNAi), CRISPR-Cas gene editing

## Abstract

*Bombyx mori* is a highly utilized model organism in immunological studies. It primarily relies on innate immunity to defend itself against bacterial or viral infections. It employs physical and molecular barriers to counter pathogens through its immune system. The viral infections of *B. mori* are suppressed mainly by RNA interference, activation of stress-response, and cell apoptosis. These roles are played by cascades as central signaling, particularly by Toll, IMD, and JAK/STAT. *B. mori* triggers humoral response, including antimicrobial peptides, as well as cellular immunity, which involves encapsulation, phagocytosis, and nodulation with the help of the haemocytes, in response to bacterial infection. Even though the pathways in bacterial and viral immunity vary, there are similarities in pathways that guarantee the orchestration of defense, which are applicable in sericulture, disease control, and comparative immunology. In addition, the knowledge of these immune mechanisms not only provides insight into interactions between insects and their hosts but also has implications in disease management and comparative immunology.

## 1. Introduction

*Bombyx mori*, the domesticated silkworm, plays a central role in the sericulture industry because it is the largest producer of silk and a significant source of life to a large section of the world population [1]. The domestication of silkworms is believed to have originated in China more than 5000 years ago and has since spread to other parts of the world [2]. Silk is a textile fiber obtained from *B. mori* that has distinctive characteristics, including tensile strength and luster, and can be dyed, which is why silk is an in-demand raw material in the textile industry [3,4]. Owing to the unique structural and functional properties of its main constituents, silk proteins, particularly fibroin and sericin, have gained prominence in a variety of advanced engineering and biomedical applications, mainly because their natural occurrence in silk facilitates biocompatibility and mechanical strength needed for their current, widespread use in reconstructive efforts targeting human tissues such as skin, bone, and nerve [5,6].

Silkworms provide a promising example in the study of insect genomics and immunology [7]. It has become a popular model organism for exploring various human health-related topics, including microbial toxicology and pathology, as they share many genetic and physiological similarities with humans and are highly studied for human antimicrobial drug screening [8,9]. Thus, the use of the silkworm as a model organism for studying human tumors, degenerative diseases, and metabolic diseases has become a current research focus. The industry’s reliance on *B. mori* ensures that silkworms are not affected by infections or other factors that could affect silk production.

However, bacterial and viral infections continue to challenge the sericulture industry, potentially causing severe economic losses. Bacterial pathogens, such as *Bacillus* species and *Streptococcus* species, from silkworms exhibiting flacherie cause severe morbidity and mortality within silkworm populations [10]. Viral infections, especially *B. mori* nucleopolyhedrovirus (BmNPV) and cytoplasmic polyhedrosis virus (CPV), are also a significant problem in sericulture [11,12]. These pathogens can spread quickly across the silkworm’s colonies and lead to a reduction in silk yield, the quality of cocoons, and, in extreme cases, cause devastating economic losses to sericulture. Due to the vulnerability of *B. mori* to these infections, a deep insight into the immune system of the silkworm and establishing effective disease control strategies are required. The silkworm acquires numerous in-built defense responses to combat its pathogens, such as physical barriers and an immune system. Physical barriers against the infection of pathogenic microorganisms can be ensured by the epidermis, tracheal respiratory organs, and midgut peritrophic membrane. Since insects do not have the adaptive immune system that vertebrates have, they depend solely on an intense and highly complex innate immune system to protect them against such infections [13].

Given the severe effects on the health of silkworm and silk production posed by bacterial and viral infections, comprehension of the immune responses activated by *B. mori* to these pathogens is essential in establishing a plan of action to protect silk production and to maintain the sericulture sector [14]. By understanding the cellular and molecular aspects of the silkworm’s immune mechanism, scientists can identify potential targets for intervention against infections and develop new strategies to prevent and control infections. Besides the traditional breeding techniques, novel biotechnological tools have emerged, such as CRISPR/Cas9 gene editing, which have changed the disease control behavior of silkworms [15]. These gene-editing systems enable targeting of viral genes or host susceptibility factors, and transgenic silkworms that express CRISPR/Cas9 exhibit significantly higher resistance to BmNPV through disruption of essential viral genes or repetitive genome sequences [16,17]. Moreover, insights into host–pathogen interaction in *B. mori* might also help shed more light on insect immunity mechanisms, and the combination of CRISPR/Cas technology with existing breeding initiatives provides potential for developing robust disease-resistant silkworm strains suitable for industrial sericulture. This review systematically covers the dynamics of the innate immune system to bacterial and viral pathogens, explaining the cellular and humoral immunity responses by which silkworms perceive pathogenic threats, implement effector responses, and trigger signal transduction pathways. Most importantly, we consider CRISPR/Cas technology a revolutionary genetic tool that enables precision-targeted interventions, either by directly disrupting viral genomes or by regulating host immune factors, providing new opportunities to acquire disease resistance in sericulture.

## 2. Background of *B. mori* Immunity

*B. mori*, lacking an adaptive immune system, operates primarily through innate mechanisms to defend against pathogenic threats [18]. This defense starts at the organismal level, with the first line of defense being the physical barriers, which consist of the cuticle, tracheal respiratory organs, and gut lining, which physically hinder the entry of pathogens (Figure 1) [19]. However, these passive barriers are insufficient to protect insects against the microbial challenges they face during development and feeding. The gut is the primary site since most pathogens invade through oral feeding, making it a frontline battleground where local immune responses are first activated [13]. Beyond these barriers, *B. mori* has developed a sophisticated active immune mechanism focused on two major tissues, the fat body and hemocytes, orchestrating systemic immune responses [20]. The fat body, analogous to the mammalian liver, serves as a principal organ for synthesizing humoral immune factors, while hemocytes circulating in the hemolymph mediate cellular defenses [21].

Humoral responses entail the release of soluble factors, e.g., antimicrobial peptides (AMPs) and melanization cascade components, into the hemolymph and the generation of reactive oxygen species [13]. Notable AMPs found in *B. mori*, which are regulated through Toll and Imd signaling pathways, are cecropin, moricin, defensin, gloverin, lebocin, and attacin [1]. Complementing these peptides, the silkworm produces one lysozyme and three lysozyme-like proteins, one of which plays a role in the eradication of invading pathogens [22]. While humoral factors are systematically distributed, cellular immunity offers a local and targeted defense system [23]. In comparison, cellular immunity consists of phagocytosis, encapsulation, nodule formation, and the induction of apoptosis in compromised cells, primarily facilitated in the hemolymph by granular cells and plasmatocytes of haemocytes [24]. These cellular responses frequently operate in conjunction with melanization, a humoral response that bridges both arms of immunity [25]. Melanization is a rapid phenoloxidase-dependent process that generates melanin and cytotoxic compounds to encapsulate and neutralize invading pathogens [26,27]. In addition to this process, apoptosis, a genetically regulated process of autonomous cell death, plays a crucial role in maintaining cellular homeostasis. Insects incapable of adaptive immunity utilize apoptosis as a principal weapon against viral defense. While research on insect innate immunity has been well-conducted, there has been no attempt to elucidate the molecular mechanisms underlying the effector systems of insect immunity [2].

Molecular studies have revealed that *B. mori* employs an innate immune system comprising pattern recognition receptors (PRRs), immune signaling cascades (JAK/STAT, Toll, and IMD), antimicrobial peptides (AMPs), RNA interference (RNAi), and cellular responses like phagocytosis and melanization to combat bacterial and viral infections [23,28]. In addition to these core mechanisms, emerging evidence demonstrates that hormonal regulation (ecdysone) [29], circadian photoperiodism [30], and gut microbiota [31] also influence immunity. Notably, Viral pathogens have adapted mechanisms to overcome these defenses, and continuous studies on the molecular immune mechanisms are required to improve disease resistance in silkworms for the benefit of sericulture.

## 3. Molecular Mechanism of Bacterial Infection in Silkworm

*B. mori* has a complex balance between molecular recognition, signal transduction, and effector responses during the innate immune response toward bacterial infection. *B. mori* mainly relies on pattern recognition receptors (PRRs), signal transduction and cascades such as Toll and IMD, antimicrobial peptide (AMP) production, and phenoloxidase (PO) promoted melanization to defend against pathogenic bacteria (Figure 2). Earlier genomic and biochemical studies have explained the immune response of silkworms to bacterial infections. The midgut is vital in combating the ingested pathogenic bacteria [32]. It has various mechanisms of immune response to different types of bacterial infections. Typically, few bacteria can survive and grow in the silkworm midgut because the intestinal pH, redox potential, digestive enzymes, and immune system collectively protect against bacterial colonization using reactive oxygen species (ROS) as a primary line of defense [33]. ROS, especially hydrogen peroxide (H_2_O_2_) and nitric oxide (NO), aid in preventing bacterial proliferation [34]. *Pseudomonas aeruginosa* and *B. bombyseptieus* infection significantly increases the levels of H_2_O_2_, NO, and ROS-related gene transcription [35]. Nevertheless, excessive reactive oxygen species production in silkworms can also have a detrimental impact on the host organism. Removal of excess ROS is carried out by antioxidant enzymes, such as superoxide dismutases, catalase, peroxiredoxins (Prxs), and NADPH oxidase [13,36,37,38]. Research discovered that oxidation resistance 1 (OXR1) shows silkworm resistance to *P. aeruginosa* via the JNK pathway [39]. Bacteria that resist ROS and remain surviving in the midgut require recognition of pathogen-associated molecular patterns (PAMPs), which include peptidoglycan (PGN) and lipopolysaccharides (LPS) of Gram-negative bacteria and lipoteichoic acids (LTAs) of Gram-positive bacteria. These patterns are exclusive to bacterial entities and are recognized by host organisms’ [40] specialized pattern recognition receptors (PRRs), including peptidoglycan recognition proteins (PGRPs) and β-glucan recognition proteins (βGRPs), initiating immune signaling cascades [41,42]. These immune signals then trigger the Toll and Imd signaling pathways, ultimately leading to the production of antimicrobial effectors [43]. For instance, *Staphylococcus aureus* and *Escherichia coli* can induce and activate the silkworm Toll signaling pathway [44,45]. Toll family member 18-Wheeler (18 W) has been observed to play a role in the immune response against bacterial infections through the targeted activation of *cecropin A* and *gloverin 2* [46]. Similarly, Toll and IMD signaling pathways ultimately suppress bacterial proliferation by stimulating the expression of antimicrobial peptides (AMPs) [1,2,23,47,48]. Recent studies demonstrated that *BmToll9-1* and *BmToll9-2* are positive regulators of the immune system in silkworms, inducing the expression of antimicrobial peptide genes after infection with *E. coli*. These Toll receptors mainly function in the midgut to maintain microbial homeostasis and maintain microbial growth [49,50]. However, further studies are required to clarify how the silkworm Toll signaling pathway responds to bacterial infection. In the hemocoel, specialist hemocytes reach the battlefield and absorb bacteria by phagocytosis. Hemocytes also promote melanization and labeling of the pathogen for destruction and prevent further spreading [51]. Prophenoloxidase (PPO) is secreted in the hindgut of the silkworm, and activation of the PPO cascade leads to blackening of the feces and reduction in the number of living bacteria at the wound site [52]. In fact, PPO cascade activation helps in the survival of argument-positive bacteria in fungi-infected fruit flies [53].

### 3.1. Pattern Recognition Receptors in Silkworm Innate Immunity

The Pattern Recognition Receptors comprised three subfamilies: peptidoglycan recognition proteins (PGRPs), β-glucan recognition proteins (βGRPs), and Lipopolysaccharide Binding Proteins (LBPs) (Table 1). In silkworms, 12 PGRPs have been identified [54]. PGRP-S1, -S2, -S3, -S4, -S5, and -S6 are short-type, all of which contain signal peptides and are secreted into the hemolymph [55,56], while PGRP-L1, -L2, -L3, -L4, -L5, and -L6 are long-type, which typically possess transmembrane domains or intracellular localization signals [55]. Studies have shown that the humoral defense mechanism of silkworms against bacterial infection includes the enhanced expression of several pattern recognition receptors (PRRs), *PGRP-S1-S3*, *PGRP-L1-L3*, *βGRP3* and *4*, which respond to both Gram-positive and Gram-negative bacteria [18,32,57]. This results in the upregulation of antimicrobial peptides, such as cecropin A and B, attacin, gloverin 2 and 3, along with ROS produced through NADPH, which collectively amplify their immune response [32,35]. Further, *BmPGRP-S4*, in particular, exhibits high expression in haemocytes and activates antimicrobial peptide genes, including *lebocin*, *moricin*, *cecropin B*, *cecropin D*, and *attacin*, as demonstrated by a dual luciferase reporter assay [58]. Recombinant *BmPGRP-S4* can bind to bacteria, inhibit their growth, and hydrolyze peptidoglycan in the presence of Zn^2+^, indicating it possesses N-acetylmuramoyl-L-alanine amidase activity [59]. Likewise, *P. aeruginosa* induces the expression of *cecropin*, *defensin*, *attacin*, *gloverin*, *lebocin*, and *moricin* in the fat body [13]. In contrast to the other short-type PGRPs, *BmPGRP-S5* functions as a negative regulator of the immune response [60]. Recent studies revealed that *BmPGRP2* exists in two isoforms, BmPGRP2-1, a transmembrane protein and BmPGRP2-2, an intracellular protein. *BmPGRP2-1* binds to DAP-type peptidoglycan and activates the canonical Imd pathway during bacterial infection [41]. Intriguingly, *BmPGRP-L4* also functions as a negative regulator of humoral responses, as demonstrated by the enhanced inhibitory effects of *E. coli* and *S. aureus* growth observed in hemolymph from *BmPGRP-L4*-silenced larvae [61]. *BmPGRP-L5* was identified in the genome-wide screen but appears to lack detailed functional characterization in bacterial infection contexts [62]. The defense mechanism of the silkworm is immune priming, which is an extraordinary feature. Repeated exposure to bacterial components, such as peptidoglycans, causes silkworms to recall the experience and produce AMP more quickly and intensely when exposed to bacteria again. This primed immunity significantly elevates the silkworm’s resistance. Additionally, AMPs are subject to negative regulation. For instance, the expression of *BmCecD*, *BmGlv2*, and *BmMoricin* in the fatbody is downregulated by *BmSerpin-15*, stimulated by *Micrococcus luteus* bacterium [63].

β-Glucan recognition proteins (βGRPs) constitute another important PRR family in silkworms [64]. Four β-glucan Recognition Proteins (βGRPs), βGRP1, βGRP2, βGRP3, and βGRP4, have been identified in *B. mori*, playing a key role in the activation of the prophenoloxidase (PPO) cascade, which is crucial for immune responses. The protein βGRP1 has a glucanase domain and a β-1,3-glucan-binding domain. It contains two active domains, a 20 kDa part of which binds to the fungal 1,3 glucans and a 43 kDa glucanase-like part activating the PPO cascade [54,60,65]. Research has demonstrated that β-1,3-glucan does not activate the PPO cascade in βGRP-deficient plasma, which is restored upon the addition of recombinant βGRP-3, indicating the essential role played by βGRP1 and βGRP3 in the activation of PPO [66]. PPO activation mechanism has been studied in other insects with β-1,3-glucans inducing self-association of βGRP-2 into a complex that binds hemolymph protease 14 (HP14) [67,68,69]. However, the exact mechanism of PPO activation in *B. mori* is not fully known yet. Research has shown that, in the process of PPO activation, HP6 and HP21 form a complex with serine protease inhibitor 5 (Serpin 5) [70]. Subsequent events need to be further investigated. There are also negative regulators of PPO activation in silkworms that have been observed. Injection of *M. luteus* and recombinant *BmSerpin-5*, *-6*, and -*15* induces a significant decrease in PO activity and deactivation of the PPO cascade by serine proteases, respectively [63,70,71].

The LBP family is important in the identification of lipopolysaccharides (LPS), which are part of the outer membrane of Gram-negative bacteria. Usually, the binding of LPS (a part of Gram-negative bacteria) by LPS-binding protein (LBP) triggers the subsequent downstream signaling. As an example, the LPS-LBP complex binds to CD14 and presents LPS to Toll-like receptors to trigger downstream signaling in humans [72,73]. In *B. mori*, LBP has also been described and suggested to be involved in the clearance of bacteria [74]. However, the exact number of LBP in silkworms has not been definitively characterized in the literature. Functional analysis has shown that BmLBPs are involved in the clearance mechanisms of bacteria in the hemolymph in case of inoculation of bacteria into the silkworm body cavity [75]. However, the mechanism of action of this protein still needs to be illustrated. *BmCTL21* (also known as BmLBP) is elevated in the fat body and carried to hemocytes, where it attaches to the surface of Gram-negative bacteria in the event of infection, and its silencing leads to suppression of PO activity [76]. Another type of pattern recognition receptor that stimulates the prophenoloxidase pathway (PPO), C-type lectin-S6 (*BmCTL-S6*), is involved in host defense against *M. luteus*, *E. coli*, and *B. subtilis*, and recombinant *BmCTL-S6* is capable of binding bacterial cell walls and modulates host melanization, encapsulation, and phenoloxidase (PO) activity. The net result of such activities is the immune response to bacterial pathogens [77].

**Table 1 insects-16-01167-t001:** Immunity-Related Proteins in *Bombyx mori*.

Protein Name	Function	Pathway	Reference
**Pattern Recognition Receptors (PRRs)**			
BmPGRP-S1, BmPGRP-S2 and BmPGRP-S3	Recognizes peptidoglycan; activates immune signaling	IMD pathway	[18,47,78]
BmPGRP-S4	Peptidoglycan binding and hydrolysis; amidase activity; promotes AMP production	IMD pathway; PPO cascade	[58,59]
BmPGRP-S5 and BmPGRP-L4	Negative regulator of AMP expression; maintains immune homeostasis	Toll and IMD pathways	[60,61]
BmPGRP-L1 and BmPGRP-L6	Broad bacterial recognition; Activates immune response to DAP-type PGN	IMD pathway	[79,80]
BmPGRP2 (BmPGRP2-1, BmPGRP2-2)	Antibacterial and antiviral immunity	IMD pathway; Antiviral	[41]
BmβGRP1 and BmβGRP3	Broad pathogen recognition; PPO activation	PPO activation cascade	[65,66]
BmβGRP4	Pathogen recognition; antiviral through apoptosis	PPO cascade; Antiviral	[81]
BmLBP	LPS recognition; opsonization; nodule formation; bacterial clearance	Toll pathway	[82,83]
BmHemolin	Pattern recognition; cell adhesion; phagocytosis	Cellular immunity	[2,84]
BmCTL-S21 and BmCTL-S6	C-type lectin; bacterial cell wall binding; promotes melanization and PO activation	PPO cascade	[76,77]
**Toll Receptors**			
BmToll9-1, BmToll9-2 and BmToll11	LPS and bacterial recognition; AMP induction	Toll pathway	[45,49,50,85]
Bm18-Wheeler	Bacterial recognition; cecropin A and gloverin 2 activation	Toll pathway	[46]
**Antimicrobial Peptides (AMPs)**			
Cecropin A, B, D, and E, Moricin, Defensin, Gloverin 1, 2, 3, and 4, Lebocin, Defensin A/B, and Attacin	Anti-Gram-negative bacteria; Anti-Gram-positive bacteria and antifungal	Toll and IMD pathways	[1,48,86,87,88,89,90]
Lysozyme	Peptidoglycan hydrolysis; direct bacterial killing	Humoral immunity	[22]
**Serine Protease Cascade Components and Serine Protease Inhibitors (Serpins)**			
BmSerpin-4	Negative regulator of the PPO cascade; regulates AMP expression	PPO cascade	[91]
BmSerpin-2,	Melanization regulation; antiviral immunity	PPO cascade; Antiviral	[92]
BmSerpin-15	Negative regulator of the PPO cascade; regulates AMP expression	PPO cascade; Toll and IMD pathways	[63]
BmSerpin-5 and 6	PPO cascade inhibition prevents excessive melanization	PPO cascade	[70,71]
Serpin-1a, Serpin-6	Serine protease inhibitors; negative regulation	Toll pathway (regulation)	[93]
SPINK7	Fungal recognition; hemocyte-mediated defense; encapsulation	Cellular immunity	[94]
BmCLIP2 and BmCLIP13	Serine protease; Spätzle processing and cuticle remodeling	Toll pathway and Developmental immunity	[20,93,95,96]
**Prophenoloxidase System**			
PPO1, PPO2 and PO	Prophenoloxidase; melanin production; pathogen encapsulation	PPO cascade	[20,96]
**Signaling Molecules and Transcription Factors**			
Paralytic Peptide (PP)	insect cytokine; immune activation; promotes phagocytosis; AMP induction	Cellular and humoral immunity	[20,26]
Relish	NF-κB-like transcription factor; IMD pathway effector; AMP transcription	IMD pathway	[47,48,80]
Dorsal	NF-κB-like transcription factor; Toll pathway effector; AMP transcription	Toll pathway	[26,97]
IMD	Immune deficiency protein; critical for IMD pathway activation	IMD pathway	[80]
Ets2	E26 transformation-specific transcription factor; represses BmRels-mediated AMP activation	Negative regulation of the IMD pathway	[97]
BmSpatzle (BmSpz1and 4)	Toll receptor ligand; immune activation	Toll pathway	[98,99]
MD2A	MD2-like protein; lipopolysaccharide recognition	Toll pathway	[100]

### 3.2. Pathogenesis of Enteric Bacteria in Silkworms

Enteric bacteria are a diverse group of microorganisms that are mostly located in the intestinal tract of various organisms. These bacteria include an amazing diversity of species, both commensal and pathogenic species. Their functions are important in host physiology, which includes nutrient processing, as well as immune system regulation. However, some enteric bacteria may be a major source of disease, especially in cases where the immune mechanisms of the host are compromised or in cases where the bacteria have certain virulent factors. Several enteric bacteria have been reported to be pathogenic that utilize a wide array of virulence factors in order to colonize, invade, and cause disease in silkworms, such as flacherie and septicemia, including *B. cereus* and *Klebsiella granulomatis* [101,102]. *B. cereus* secretes casease, lipase, and amylase, and this contributes to its pathogenicity [101]. Likewise, *Serratia marcescens* synthesizes chitinases, which degrade chitin, a major constituent of the insect exoskeleton and peritrophic matrix [103]. Bacteria such as *Staphylococcus* sp., *B. thuringiensis*, *Enterococcus mundtii*, and *Serratia marcescens* have also been identified as the disease-causing agents of the silkworms, indicating their potential role in silkworms [104]. Certain bacteria are able to evade phagocytosis by haemocytes or inhibit the synthesis of AMPs. Similarly, *E. coli* mutants enhance the release of vesicles that exhibit resistance to different antibiotics and antimicrobial peptides [105]. The capability to evade the silkworm’s immune system is a critical determinant of bacterial pathogenicity.

Intestinal microflora of silkworms has multifaceted effects on health and vulnerability to enteric infection. Whereas certain commensal bacteria can offer to withstand pathogens because they compete for resources or produce inhibitory compounds, others can lead to infection or disease [106]. Silkworms can be subjected to transient high temperature, which can modify the intestinal flora and induce modifications in immune-related genes [107]. Further research is needed to obtain a complete overview of the interactions between the silkworm gut microbiota and enteric pathogens.

### 3.3. Wound-Induced Antibacterial Mechanism in Silkworms

The wound-induced antibacterial mechanisms in the silkworm *B. mori* entail a number of innate immune responses activated when the silkworm is injured to guard against pathogens. Silkworms frequently face wound infections in their rearing environments, where bacteria can enter through these wounds and invade the insect’s internal body cavity, called the hemocoel. Utilizing direct inoculation of bacteria into the silkworm hemocoel facilitates research on immune recognition, signaling molecules, and bacterial virulence factors, aiding in understanding host–pathogen interactions [108]. Hemocytes are pivotal in the defense against pathogenic bacteria within the circulatory system of silkworms, serving as a primary site for immune responses [109]. Cellular immunity is predominantly controlled by two key factors. Firstly, it involves the elimination of pathogenic microorganisms through plasmatocytes. Secondly, protease cleavage primarily induces haemocyte agglutination and melanin deposition. A major wound healing and antibacterial reaction of silkworm hemolymph is melanization. The process is facilitated by phenoloxidases, and transglutaminase-catalyzed cross-linking stabilizes the immune complexes. It is inhibited through a parasitoid wasp-derived melanization inhibitor which is known as Egf1.0, showing its significance in immune defense [110]. Phagocytosis involves bacterial clearance by an individual hemocyte cell. Cells initially adhere to bacteria before altering the cytoskeleton to internalize and eliminate bacteria via phagocytosis. Insect fat bodies can analogously be likened to the liver in mammals [111]. Antimicrobial peptides (AMPs) can be endogenously synthesized by adipocytes and subsequently released into the hemolymph of *B. mori*. Introduction of bacteria intracellularly induces adipocyte activation, the production, and secretion of AMPs [80]. *BmPGRP-L4* regulates the expression of AMP in a manner that ensures a state of immune homeostasis through negative feedback repression to prevent overactivation of the immune system [61]. Also, *Bmponericin-L1* exhibits antibacterial and anti-biofilm action, via membrane destabilization of *P. aeruginosa* [112].

In summary, the wound-induced antibacterial response in *B. mori* entails an organized innate immune response, which comprises immune complexes, melanization, AMP synthesis, and cellular immune responses. These collectively act to prevent infection and help in wound healing.

## 4. Viral Infections in Silkworms

Viral infections in silkworms, particularly those impacting *B. mori*, pose significant challenges in sericulture, primarily due to their effects on silk yield and quality. Efforts have been made to develop effective treatments for viral diseases in silkworms, but complete eradication of the viruses and overcoming the problem entirely remains a challenge [113,114]. Among the silkworm diseases, Viral infections account for about 80 percent of total cocoon loss, constituting the highest percentage of total losses in sericulture [115]. These infections are caused by *B. mori* nucleopolyhedrovirus (BmNPV), cytoplasmic polyhedrosis virus (BmCPV), densonucleosis virus (BmDNV), infectious flacherie virus (BmIFV), and bidensovirus (BmBDV), being particularly significant [116]. Silkworm viral infections are classified as inclusion-type or non-inclusion viruses. Inclusion viruses such as *B. mori* nucleopolyhedrovirus (BmNPV) and cytoplasmic polyhedrosis virus (BmCPV) produce polyhedral inclusion bodies (PIBs) readily observable under standard microscopy. Non-inclusion type viruses comprise infectious flacherie virus (BmIFV) and densonucleosis viruses (DNVs), which are detectable through multiple complementary methods, including serological assays, electron microscopy, and immunofluorescence analysis [117]. Viral infections are usually caused by silkworms consuming contaminated food, which causes viruses to enter the midgut of the silkworms. The effect of this is degradation of the plasma membrane followed by penetration into the hemocoel [118].

### 4.1. Major Immune Molecules and Immune Responses of B. mori to Viruses

The interplay between viruses and their hosts is a dynamic process, with the virus attempting to replicate and spread, and the host’s immune system responding to control and eliminate the infection. *B. mori* exhibits a sophisticated and complex inborn immune reaction to viral infections, utilizing a variety of immune molecules and pathways. The innate immune system relies on different PRRs and signaling molecules. BmPGRPs and C-Type Lectins. BmPGRP are more specific receptors that are expressed on cells of the innate immune system with the ability to detect particular molecular patterns common to pathogens, such as viruses [119]. C-type Lectins target glycan substructure displayed on the envelope of the virus. When they bind, they can take up viral particles and destroy them, initiating innate immunity—namely, phagocytosis, antigen processing, and presentation—which, in turn, activate T cells and the adaptive immune system [120]. In addition to these classical immune pathways, the midgut is a key first-line immune defense mechanism in silkworms, serving as a physical and immunological barrier against pathogen invasion [121]. The midgut consists of specific cell types and secretes midgut juice, a blend of digestive enzymes and antimicrobial factors that play a significant role in the insect’s immune defense [122]. Recent proteomic research has identified a novel digestive proteinase, lipase member H-A, that contributes to digestive juice antiviral activity against BmNPV [7]. Also, digestive proteinase, such as trypsin, has been shown to play an antiviral role, and its expression is highly upregulated following BmNPV infection [123].

The broader antiviral immune response in silkworms involves the coordinated control of multiple defense mechanisms. These include the antiviral immune response in silkworms must involves the control of different defense mechanisms, such as antimicrobial peptides (AMPs), reactive oxygen species (ROS), RNA interference (RNAi) pathway, the stimulator of interferon genes (STING) pathway, as well as the Toll, Imd, and JAK/STAT pathways (Figure 3) [118,124,125].

#### 4.1.1. RNAi-Mediated Antiviral Responses and Suppressors of RNAi Encoded by BmNPV

Insects use three types of small RNAs: small interfering RNAs (siRNAs), microRNAs (miRNAs), and PIWI-interacting RNAs (piRNAs), which play key roles in gene regulation and immunity. The canonical RNAi pathway begins with the recognition of double-stranded RNA (dsRNA), which is processed by the enzyme Dicer-2 into short siRNAs. These siRNAs are loaded onto the Argonaute protein (especially BmAgo2 in silkworm), forming the RNA-induced silencing complex (RISC). siRNA is used by the RISC as a guide to bind to complementary mRNA, leading to its cleavage and degradation, thereby silencing the target gene expression [126]. The significance of RNA interference in combating viruses like BmNPV and BmCPV is fully understood [127]. Silkworm viruses did not induce the expression of *Dicer2* and *Ago2*; however, deep sequencing results showed that an abundance of viral siRNA (~20 nucleotides) was produced in infected insect hosts of baculovirus and BmCPV, suggesting that RNAi response is one of the prominent antiviral defenses of hosts [128]. The RNAi response, utilizing *Dicer-2*, has been shown to degrade transcripts of *Helicoverpa armigera* single nucleopolyhedrovirus (HaSNPV), cleaving double-stranded RNA (dsRNA) genomes during viral replication into vsiRNAs [129]. Similarly, upon infection with BmNPV, *Dcr2* expression is markedly upregulated, which suggests its direct involvement in mounting an RNAi response against viral invasion [130]. Although the main RNAi machinery comprises *Dcr2* and *Ago2*, accessory proteins like R2D2 and Translin are needed during *Drosophila* to sort siRNAs into Ago2-containing RISCs. But in *B. mori*, the studies of the Bm5 cell line and different tissues of the silkworm display considerably low expression of *R2D2*. On the same note, *Translin* mRNA is mostly deficient in Bm5 cells [131]. These molecular characteristics suggest that inherent deficiencies or low abundance of such accessory proteins could partially explain the variable sensitivity and robustness of RNAi responses in *B. mori*.

Viral suppressors of RNA interference (VSRs) have been developed in many silkworm viruses, such as the p35 protein, which interferes with siRNA production or function, consequently enhancing the expression of crucial viral genes that promote replication [132]. BmCPV NSP8 has been shown to interact directly with *BmAgo2*, preventing the efficient cleavage of viral RNAs by Ago2, promoting viral proliferation in host cells [127]. Further, BmNPV expresses a microRNA, BmNPV-miR-1, which activates and silences the host protein Ran and represses host miRNA-signaling, which is a part of antiviral defense mechanisms [133], and BmCPV-miR-1 might mediate target gene *BmIAP* expression and BmCPV replication [134]. Moreover, viral proliferation can be suppressed by host miRNA. As an illustration, the *ie-1* gene of BmNPV can be silenced by bmo-miR-2819, which suppresses viral replication; however, bmo-miR-278-3p may reduce target gene *IBP2* expression and raise BmCPV mRNA levels, as bmo-miR-278-3p is repressed and *IBP2* is elevated in BmCPV-infected silkworms [128]. Additionally, recent investigations have revealed that miRNA, bmo-miR-6498-5p, suppresses BmNPV infection by downregulating *BmPLPP2* to modulate pyridoxal phosphate (PLP) content in *B. mori* [135]. Similarly, bmo-miR-3351 can regulate the levels of glutathione and prevent BmNPV growth by downregulating *BmGSTe6* in *B. mori* [136]. Hence, the investigation of miRNAs involved in host–virus interactions holds significance for future research.

#### 4.1.2. JAK/STAT Pathway Modulation by Host and Virus

JAK/STAT is a crucial mechanism involved in numerous biological functions, such as immune regulation and cell growth in insects [137,138]. Vital genes of the JAK/STAT pathway, including *DOME*, *HOP*, *STAT-S*, and *STAT-L*, have been found. Studies have revealed that this pathway is significant in cell proliferation and the development of wing disks in silkworms [139]. When *B. mori* is confronted by BmNPV, the JAK-STAT pathway is a vital element of the defensive arsenal of the host. Viral infection upregulates the expression of both *BmSTAT-S* and *BmSTAT-L*, the two splice variants of the silkworm’s *STAT* gene. RNAi-mediated knockdown of either *BmSTAT* variant in cultured cells or transgenic silkworms results in significantly increased viral gene expression and enhanced BmNPV replication, demonstrating that *STAT* factors are essential for mounting effective antiviral immunity [140]. In contrast to BmCPV, Liu et al. reported that BmNPV and BmBDV induce the expression of *BmSTAT* in silkworms, suggesting that the JAK/STAT pathway might be triggered by the DNA viruses in silkworms [137]. These investigations suggest that the JAK/STAT pathway of *B. mori* did not respond to RNA virus BmCPV and can be triggered by DNA viruses. Further, heat shock protein 90 (Hsp90) inhibition can suppress BmNPV proliferation in *B. mori,* which in turn triggers the upregulation of *STAT* and suppressor of cytokine signaling protein 2 (SOCS2) and SOCS6 downregulation, indicating the possible cooperation of Hsp90 in viral resistance by activating the JAK/STAT pathway [141]. Similarly, BmHSP90 (heat shock protein) by 17-AAG and lysine acetylation alterations inhibit viral replication, which reveals new viral features of host defense that can be used to improve sericulture [142]. However, the involvement of the *BmSTAT* key transcription factor in the JAK/STAT pathway in the BmNPV infection and its pathways has not been reported yet.

JAK/STAT-specific host cytokines, *BmVagolike*, and their overexpression demonstrate the increased antiviral resistance and reduced viral loads. Conversely, reduced expression of *BmVagolike* weakens the antiviral response, supporting once again the role of cytokine-mediated JAK/STAT activation [128,143]. Thus, an enhancement of the JAK/STAT response through genetic/molecular manipulations is suggested as one of the effective methods of enhancing silkworm resistance in practical applications.

#### 4.1.3. Toll and IMD Pathway Responses to Viral Infection

The Toll and IMD pathways are essential in the identification of pathogens and triggering downstream defenses. These pathways result in the expression of antimicrobial peptides (AMPs)-controlled genes, which are regulated by two NF-kB transcription factor orthologs [144,145] and other immune factors that help to clear the infection. The relevance of the Toll pathway on antiviral defense in *B. mori* has been shown by several studies. *B. mori* is associated with augmented expression of antiviral RNAi machinery, alongside suppressed viral replication [146]. *B. mori* reverse transcriptase mediates the conversion of BmCPV, a double-stranded RNA virus, into circular DNA (vcDNA-S7), which serves as a template for generating antiviral small interfering RNAs (siRNAs) that suppress BmCPV replication both in vitro and in vivo [147]. Zhao et al. showed that the overexpression of the peptidoglycan recognition protein S2 (*BmPGRP-S2*) in transgenic silkworms dramatically improved antiviral immunity against *Bombyx mori* cytoplasmic polyhedrosis virus (BmCPV), indicating that *BmPGRP-S2* is vital in the antiviral defense mechanisms of the host [145]. Further research indicated that *BmPGRP-S2* was an extracellular protein, possibly perceiving some viral element and then signaling to the downstream molecules, and its upregulation increased *BmImd*, *BmRelish*, and AMPs expression and decreased silkworm mortality after BmCPV infection [145]. Further, *BmPGRP2-2* expression negatively regulates PTEN, promoting PI3K/Akt signaling, inhibiting apoptosis after BmNPV [41]. Overexpression of *BmβGRP4* promotes BmNPV-induced cellular apoptosis and positively regulates *BmPTEN* while negatively regulating *BmIAP* [81]. In *Drosophila*, Imd and Toll pathways have been shown to contribute to antiviral immunity [148]. Some evidence indicates that AMPs have antiviral effects in *Drosophila*, although the mechanisms of this antiviral activity remain largely unstudied. Thus, further detailed investigations are required [149]. Research on the *D. melanogaster* has shown that *Toll-9* enhances the expression of RNA interference (RNAi) elements, which is a key antiviral pathway of insects [146]. Taken together, these outcomes contribute to the knowledge of insect antiviral immunity in *B. mori* and provide the foundation of future studies and practice in sericulture health management.

#### 4.1.4. STING and Antiviral Immunity in *B. mori*

The cGAS-STING pathway is the DNA-sensing and IFN-inducing pathway, which is highly conserved in mammals and also found in arthropods, including *B. mori* and *Drosophila*. STING serves as a downstream mediator of antiviral responses in these insects, but through interferon-independent pathways. Research with *Drosophila* indicates that STING activation results in NF-κB-regulated gene expression and provides antiviral protection [150,151,152]. Although there is a paucity of direct research on the STING pathway in *B. mori*, similar innate immune pathways that overlap or interact with STING signaling have been reported. For instance, Sirtuin 5, an (NAD^+^)-dependent histone deacetylase, suppresses BmNPV replication by activating Relish, a major transcription factor regulated by NF-κB signatures associated with STING signaling [153]. STING induces antiviral immunity against BmNPV in silkworms by facilitating NF-kB activation [154]. During BmNPV infection, synthesis of cyclic guanosine monophosphate–adenosine monophosphate (cGAMP) is initiated, leading to the activation of *BmSTING*, which processes *BmRelish*. Subsequently, the stimulated *BmRelish* is carried to the nucleus to initiate the transcription of AMPs [154]. This suggests that the STING-dependent NF-κB-mediated antiviral responses are functionally relevant in *B. mori*.

Advancing studies revealing novel cyclic dinucleotides and STING agonists that activate broad antiviral programs in insects, positioning *B. mori* as a valuable model to uncover unique antiviral immunity pathways relevant for controlling viral diseases in sericulture [155,156].

#### 4.1.5. Apoptosis as an Antiviral Defense

Apoptosis is a biologically regulated process of cellular demise, and is considered a crucial component of the animal immune response to viral infections [157]. Certain lepidopteran species have been shown to defend against baculovirus infection by selectively inducing apoptosis in infected midgut epithelium cells, leading to the shedding of the affected cells [158]. *B. mori* protease-activating factor-1 (*Bmapaf-1*) activates caspases *B. mori* caspase-1 (*Bmcas-1*) and *Bmnedd2*-like caspase and controls the apoptosis of infected cells, hence minimizing BmNPV viral infection [159]. Overexpression of *Bmcaspase-1* and *B. mori* Cytochrome c (*Bmcytc*) inhibits BmNPV viral infection by upregulating apoptosis, while its knockdown promotes viral proliferation [160,161]. Moreover, the hematological and neurological expressed 1-like protein (HN1L) enhances cell survivability, decreases fragmentation of DNA, and inhibits expression of Bax protein and the activation of *caspase-9*, suggesting that HN1L is an anti-apoptotic protein in the presence of a virus [162]. In addition, C-type lectin (*BmIML-2)* triggers the activities of BmNPV-induced apoptosis and inhibits viral replication, indicating that immune recognition proteins have the ability to induce apoptosis to fight viral infection [163]. Likewise, post-transcriptional modification and deacetylation of *BmAda3*, a protein involved in chromatin, stabilized proapoptotic factor P53, resulting in elevated apoptosis during BmNPV infection and decreased viral growth [164]. Recent research indicates that Tetrapanin gene *BmTsp.C* is upregulated upon viral infection, lowers the apoptotic cell count and enhances the growth of BmNPV, and its knockdown inhibits viral proliferation [165].

To explore the apoptosis-associated response involved in BmNPV infecting silkworms. Different transcriptomics studies have shown that the apoptosis-related genes are significantly expressed in the resistant silkworm strains under AcMNPV challenge and enriched the apoptosis pathways in the hemolymph of virus-resistant silkworms [166]. Long non-coding RNAs (lncRNAs) are associated with the mitochondrial apoptosis pathway; specifically, the overexpression of *LINC5438* enhanced the proliferation of BmNPV, whereas its suppression inhibited its proliferation, indicating that *LINC5438* plays an important role in BmNPV infection [167]. Such insights into the molecular mechanisms of apoptosis in *B. mori* will offer a platform upon which the breeding of virus-resistant silkworms and the development of antiviral measures can be built.

## 5. Distinct Pathways in Silkworm Immunity Against Bacterial and Viral Infection

The innate immune system in silkworm *B. mori* is triggered by bacterial and viral infections as part of the initial defense mechanism. This defense is based on PRRs, which are used to discover molecular patterns that are conserved and linked to pathogens and induce downstream signaling pathways and effector mechanisms. Despite some of the similarities, *B. mori* is widely different in the way it recognizes and responds to viral and bacterial threats (Figure 4).

### 5.1. Extracellular Versus Intracellular Sensing

Extracellular pattern recognition receptors PRRs in *B. mori* are the primary receptors that detect bacterial infection in response to conventionally preserved pathogen-associated molecular patterns (PAMPs) present on bacterial cell walls. Gram-positive and Gram-negative bacteria have peptidoglycan (PGN) in their PAMPs, and Gram-negative bacteria consist of lipopolysaccharides (LPS) [58,168]. Concurrently, Gram-negative binding proteins (GNBPs), which bind to LPS, activate downstream immune signaling and also help upregulate innate immune genes through NF-κB transcription factors [168]. The *BmPGRP2* exists in two forms: *BmPGRP2-1* and *BmPGRP2-2*; it has been noted that *BmPGRP2-1* binds to (DAP)-type peptidoglycan and triggers the immune deficiency (Imd) response [41].

By contrast, viral recognition in *B. mori* is not well-studied in comparison with bacteria, but occurs mostly through intracellular signals mediated by PRRs, some Toll-like receptors (TLRs), and RNAi components implicated as viral sensors. For instance, recent studies have revealed that overexpression of *β-1,3-Glucan Recognition Proteins (GRPs) βGRP*, such as *βGRP-3*, hinders viral proliferation, whereas knockdown enhances it, indicating a direct antiviral role [169]. Similarly, *BmβGRP-4* inhibits the proliferation of BmNPV by inducing apoptosis [81]. In *Drosophila*, *Toll-9* is able to identify viral double-stranded RNA (dsRNA) in endosomes, triggering antiviral RNA interference (RNAi) pathways such as *Dicer2* [146]. Although direct evidence of *B. mori* is limited, the presence of toll receptors suggests that a similar mechanism may be involved. Furthermore, RNA interference (RNAi) or apoptosis regulation is caused by viruses BmNPV and BmCPV. *BmPGRP 2-2* is induced during viral infection and modulates host cell apoptosis through the PTEN-PI3K/Akt signaling pathway to favor viral replication [41]. Additionally, in the fatbody of silkworm, C-type lectin *BmIML-2* was highly expressed and strongly induced following BmNPV infection, which induces apoptosis [163]. Additionally, the BmNPV virus encodes microRNAs that regulate host gene expression to cause viral proliferation [14]. This essential distinction corresponds to a difference in the life cycles and localizations of cells by bacteria and viruses.

### 5.2. Distinct Bacterial and Viral-Induced Immune Pathways

Toll and IMD pathways are the major immune pathways that are activated by bacterial infection. Toll pathway is more receptive to Gram-positive bacteria and fungi, whereas the Imd pathway is receptive to Gram-negative bacteria [26,170]. These pathways result in the production and release of antimicrobial peptides (AMPs) that include cecropin, defensin, moricin, gloverin, attacin, and lebocin, which are useful in the attack on bacteria in the hemolymph [1]. In *B. mori*, Toll receptors like *BmToll11* and *BmToll9-1* are activated by Spatzle proteins (BmSpz2) upon pathogen recognition, inducing expression of AMPs [45]. Further, *BmPGRP2-1* induces the Imd pathway against bacteria [41]. Cellular immune responses, including phagocytosis by haemocytes, also contribute to bacterial clearance [26]. Recognition proteins, such as PGRPs and GNBPs, elicit signaling cascades leading to regulated NF-κB family transcription factors and the expression of AMP genes [47]. Another significant defense response to bacterial infection is melanization, mediated through the prophenoloxidase (PPO) activation system to give rise to phenoloxidase (PO) activity. The process leads to the formation of melanin, which surrounds and destroys pathogens [2,92]. Additionally, the immune responses that are stimulated by 20-hydroxyecdysone through hormonal regulation also include AMP expression to bacteria [100].

Comparatively, viral immune response in *B. mori* engages distinct and additional mechanisms. RNA interference (RNAi) acts as a key antiviral pathway. Specifically, the Dicer-2 (Dcr2) enzyme cleaves viral dsRNA into small interfering RNAs (siRNAs), which instruct the process of destroying viral RNA, thereby repressing virus replication [130]. For instance, *Dcr2* mRNA expression is significantly induced after BmNPV infection in the midgut and haemocytes [130]. Also, the Janus kinase/signal transducer and activator of transcription (JAK/STAT) pathway contributes to antiviral responses. Two splicing isoforms of the *BmSTAT* gene regulate resistance to BmNPV infection by controlling viral gene expression and survival rates in infected silkworms [140]. Furthermore, Apoptosis regulation is also crucial, as viruses manipulate host cell death pathways to prolong infected cell survival; *BmPGRP2-2*’s modulation of PTEN-PI3K/Akt signaling exemplifies this viral immune evasion strategy [41]. Viral proteins, including viral-encoded miRNAs, manipulate the host immune system at the transcriptional and post-transcriptional levels to evade host defenses [14]. Viral infections also cause downregulation of certain metabolic and immune-related genes, indicating that viruses can suppress host immunity to facilitate their replication [171]. These distinctions underscore that bacterial clearance relies heavily on canonical innate immune signaling and AMP production, whereas antiviral defense prioritizes RNAi-based genome surveillance and modulation of apoptotic pathways.

### 5.3. Modulation and Metabolic Adaptations by Pathogens

The viruses and bacteria have developed ways of circumventing or subduing the *B. mori* immunity. Like, AMP expression is negatively regulated by *BmPGPR-L4* to achieve immunological homeostasis [61]. Being a negative regulator, *BmPGPR-L4* has been suggested to mediate the feedback regulation of the immune signaling pathways of the silkworm to avoid extreme activation of the immune response.

Viruses can modulate host signaling pathways to evade immune defenses. Indicatively, BmNPV induces *BmPGRP2-2* expression, which suppresses apoptosis by negatively regulating PTEN and activating the PI3K/Akt pathway [41]. Likewise, BmCPV codes BmCPVmiR10, which suppresses the expression of *BmCSDE1* and *BmApaf1*, facilitating viral replication [172]. Additionally, the immune response of *B. mori* also depends on metabolic adaptations, which are achieved by generating an adenosine triphosphate (ATP) in response to BmNPV infection, which supplies energy to generate BmNPV resistance [173]. Correspondingly, the aryl hydrocarbon receptor (AhR) induced by Tryptophan metabolism via the exogenous virus BmNPV triggers the silkworm’s immune system to overcome BmNPV infections [173]. Understanding the molecular pathways will empower researchers and breeders in sericulture with potential options to improve disease resistance through genetic engineering, including CRISPR/Cas9-mediated immune boosting or RNAi-mediated antiviral therapy [19]. This knowledge enables the design of robust silkworm strains, reducing losses and improving silk productivity.

### 5.4. Similarity Effector Molecules and Immune Outcomes

In *B. mori*, antimicrobial peptides (AMPs) serve as the central effector molecules against bacterial infections. These peptides function by targeting bacterial membranes and inhibiting growth. The profile and potency of AMPs vary among silkworm strains, correlating with differing bacterial sensitivities. AMP production is essentially fat body–derived and secreted into hemolymph, where they exert their bactericidal functions [1,170]. *B. mori* produces a variety of AMPs, including cecropin, defensin, moricin, gloverin, attacin, and lebocin, which are induced by the Toll and Imd pathways [1].

Conversely, the defense mechanisms against viral infections in *B. mori* pivot largely on the effector molecules, including components of RNAi-related genes, regulation of apoptosis, and expression of specific antiviral proteins such as seroins—small silk gland-derived proteins with inhibitory effects on baculovirus [41,174]. The immune response to viruses often involves modulation of host gene expression networks rather than synthesis of AMPs as primary effectors, but AMPs like lysozyme and lebocin are induced upon viral infection and exhibit antiviral activity and are also found in haemocytes of silkworm larvae, indicating their role in systemic antiviral defense [175,176]. Moreover, melanization responses may be downregulated or modulated by serpins during viral infection to balance immunity and host survival, such as *Bmserpin2*, which regulates melanization during viral infection and may be suppressed to modulate immune responses [92]. Also, *Bmserpin3* has been shown to interact with serine protease 7 to regulate prophenoloxidase activation and modulate immune responses during viral infection [177]. The coordinated regulation of these defense mechanisms allows silkworms to mount targeted antiviral responses while maintaining physiological homeostasis.

## 6. CRISPR/Cas9 Applications in *B. mori* Immunity: Antiviral and Antibacterial Strategies

Clustered Regularly Interspaced Short Palindromic Repeats-associated (CRISPR-Cas) system, a cutting-edge genome editing tool, has rapidly transformed various fields within the natural sciences, showcasing significant advancements in a brief timeframe. Considerable advancements have been achieved across disciplines like entomology and biotechnology, with *B. mori* playing a critical role in sericulture and various scientific domains. The CRISPR-Cas9 genomic editing technology is commonly utilized for genetic investigations, enhancing pathogen resistance, and as an imaging tool in *B. mori* studies [15].

Recent research has focused on optimizing CRISPR-mediated modifications in *B. mori* to enhance traits such as disease resistance [178]. The initial effective alteration of the genome of *B. mori* using the CRISPR-Cas9 system was reported [15], where authors targeted an essential gene, *BmBlos2*, that is orthologous to the *Blos2* human gene. The successful manipulation of this gene highlighted the feasibility of using CRISPR-Cas9 not only in *B. mori* but also in other lepidopteran insects and revealed the CRISPR-Cas9 system’s applicability in pest control approaches. In *B. mori*, CRISPR technology can target key innate immune components such as peptidoglycan recognition protein (PGRP), antimicrobial peptides (AMPs), or antiviral effector like seroins to study their precise roles and to engineer enhanced immunity against specific pathogens [174]. Current progress and emerging strategies focus on antiviral and broader immune engineering.

### 6.1. CRISPR/Cas9 Targeting BmNPV for Enhanced Host Resistance

Recent studies have utilized the CRISPR/Cas9 systems to shred the BmNPV genome at repetitive viral sequences, thus preventing its capacity to replicate and repair itself. This method produces BmNPV-resistant transgenic silkworms with robust and broad resistance, demonstrating considerable promise for industrial-scale sericulture [16]. Similarly, constitutive expression of the guide RNAs and Cas9 in silkworms, silencing important viral genes including *ie-1* and *me53*, imparted inherited antiviral resistance in the offspring, providing a stable, long-term solution to viral outbreaks [179]. Additionally, to minimize developmental side effects, virus-activated promoters are used to induce Cas9 only at the time of infection, resulting in reduced off-target effects and maintaining normal host growth and development [17]. Multiplex CRISPR/Cas9 technologies can target a large number of important viral genes at once and hinder viral replication by editing genes like *ie-1*, *gp64*, *lef-11*, and *dnapol*, which increases the silkworm’s antiviral capacity [180]. Silencing of *lef8* and *lef9* showed marked resistance and reduced virus levels after CRISPR-mediated gene disruption [181]. Also, *BmSPP* negatively regulates antiviral immunity, and its CRISPR/Cas9-mediated knockout decreased BmNPV proliferation and improved silkworm survival rates, making it a promising resistance target [182]. Additionally, beyond Cas9, CRISPR-Cas12a (Cpf1) systems were shown to be even more efficient than Cas9 at conferring viral resistance, expanding the range of available genome engineering tools [183].

### 6.2. CRISPR/Cas9 in Antibacterial Immunity and Broader Immune Modulation

CRISPR/Cas9 is also essential in the functional analysis of silkworm immune genes. Knockout of *BmHemolin* revealed roles in activating cellular immunity (hemocyte melanization, aggregation) and controlling antimicrobial peptide levels for humoral defense [184]. The Kunitz-type protease inhibitor *KPI5* knockdown via CRISPR/Cas9 regulates antibacterial peptide gene expression and hemolymph melanization, confirming its dual regulatory function in immune homeostasis [185]. Similarly, Knockout studies of *sericin 1* using CRISPR/Cas9 alter the expression of immune genes and protease inhibitors in the silk gland, highlighting an indirect but significant contribution to the larval antibacterial response [186]. CRISPR/Cas9-driven knockouts of serpins and related cascade proteins have shown their adverse regulatory effects over the Toll pathway, central to antibacterial defense and immune homeostasis. This includes regulating pro-Spätzle processing and AMP production in response to bacterial challenge [93]. CRISPR/Cas9 tools have also been applied to genetically modify bacteria isolated from the *B. mori* gut, such as *Pseudomonas fulva*. These bacteria may be involved in silkworm gut immunity and pathogen resistance, and the genetic tools allow further insight into the bacteria–silkworm interactions and antibacterial action in the holobiont environment [187]. High-throughput CRISPR/Cas9 screening in *B. mori* can scale up to the discovery of large numbers of antibacterial effectors and regulators in a functional screen.

CRISPR/Cas9 technologies have revolutionized genetic research and immune engineering in *B. mori*, enabling targeted, stable, and practical genetic improvements against viral and bacterial threats as well as facilitating broader biomedical applications. Such breakthroughs open the way to improved agricultural productivity and strong in vivo research platforms in immunology and pharmaceutical development.

## 7. Conclusions

This paper reviews the innate immunity of the *B. mori* defense system tailored to combat bacterial and viral infections, mainly based on pattern recognition receptors, antimicrobial peptide-based humoral effectors, and cellular responses like phagocytosis and melanization. The distinct immune response to bacteria and viruses underlines how the silkworm differentiates extracellular bacterial constituents via Toll and Imd signaling pathways and uses intracellular antiviral programs, including RNAi and apoptosis regulation, to restrain viral growth. Progress on regulatory networks, such as the functions of the JAK/STAT and STING signaling pathways, provides the mechanistic basis for targeted interventions in silkworms for health management.

Furthermore, the prospect of the application of genome editing tools and, in particular, CRISPR-Cas systems can provide new opportunities to make *B. mori* more resistant to diseases by exact regulation of the immune response-related genes. Biotechnological applications can not only facilitate unraveling gene functions but also enable the engineering of silkworm strains that are more resilient to pathogens, which is the most important factor in maintaining sericulture productivity and reducing economic losses. Further studies on the topic are required to validate, using functions, the use of candidate immune modulators and to develop synergistic strategies of genetic, molecular, and environmental factors that can result in strong and sustainable immunity in silkworms.

Collectively, an enhanced molecular understanding of *B. mori* immunity against bacterial and viral infection, with novel genome editing methods, preconditions a revolutionary breakthrough in the management of the diseases in sericulture. This will not only protect silk production but also benefit the general field of insect immunology and biotechnology, with its goals possibly going beyond sericulture to the pest control of ecosystems and agriculture.

## Figures and Tables

**Figure 1 insects-16-01167-f001:**
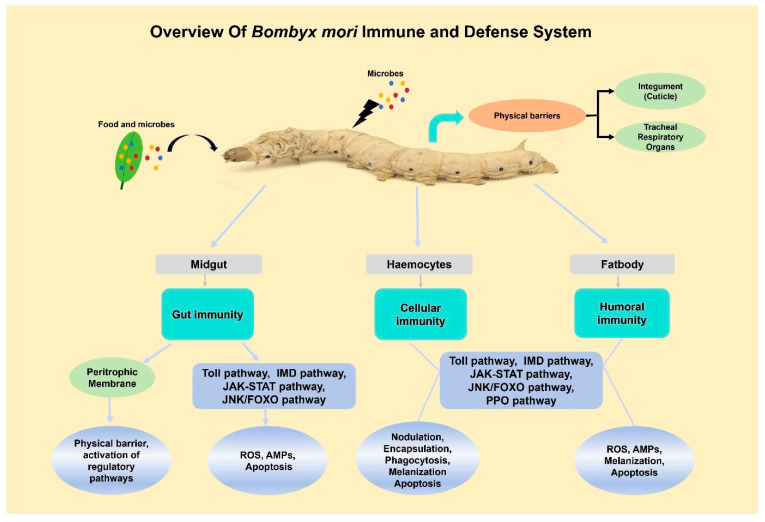
Overview of *B. mori* immune system highlights physical barriers, active paralytic peptide (PP) response, and important components used in cellular and humoral immunity.

**Figure 2 insects-16-01167-f002:**
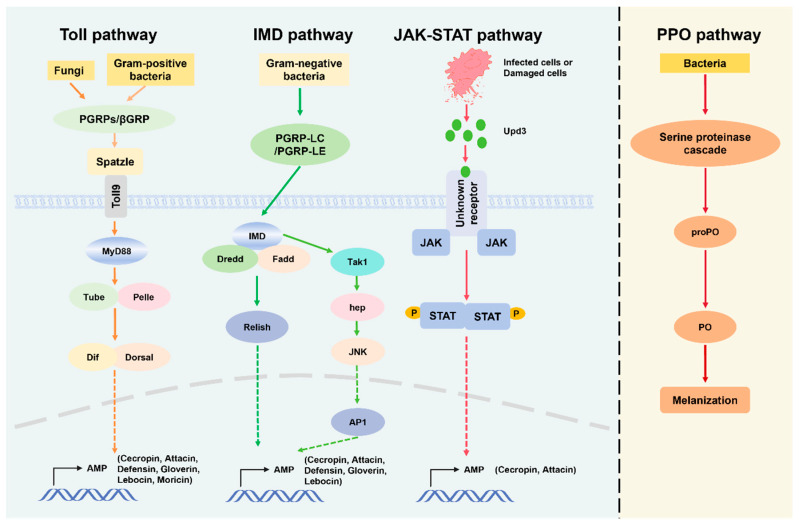
Illustrating the Toll, IMD, JAK-STAT, and PPO immune response pathways in a host organism. The Toll pathway is activated by Gram-positive bacteria and fungi, involves PGRPs/βGRP interacting with Spatzle, activating MyD88, and subsequent AMP gene expression. IMD pathway, triggered by Gram-negative bacteria, uses PGRP-LC/PGRP-LE to activate IMD, which in turn activates Dredd and Fadd, culminating in the release of antimicrobial peptides (AMPs). The JAK-STAT pathway responds to infected or damaged cells, with Upd3 binding to the receptor to activate JAK and STAT, promoting AMP synthesis. The PPO pathway is involved in the melanization response, initiated by bacteria and activating a cascade of serine proteinases that leads to the conversion of proPO to PO, facilitating melanization.

**Figure 3 insects-16-01167-f003:**
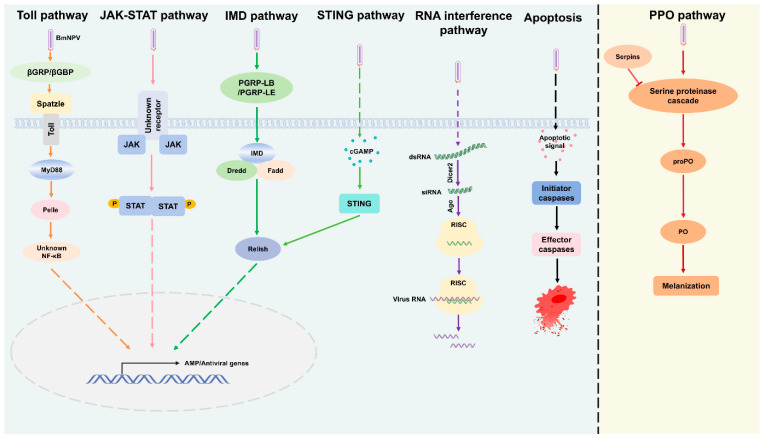
Overview of antiviral immune response pathways in the host, including the Toll, JAK-STAT, IMD, STING, RNA interference, Apoptosis, and PPO pathways. These pathways regulate immune defense mechanisms against viral and microbial infections. The Toll pathway is triggered by βGRP/βGBP, while the JAK-STAT pathway mediates cytokine signaling for AMP/Antiviral gene expression. The IMD pathway, activated by PGPR-LB/PGPR-LE, regulates immune responses via Relish. The STING pathway, activated by cGAMP, defends against viral DNA. The RNA interference pathway targets viral RNA using dsRNA, siRNA, and RISC. Apoptosis signaling removes infected cells through caspase activation. Finally, the PPO pathway, through a serine protease cascade, aids in immune response via melanization. These pathways work together to coordinate immune activation and pathogen clearance.

**Figure 4 insects-16-01167-f004:**
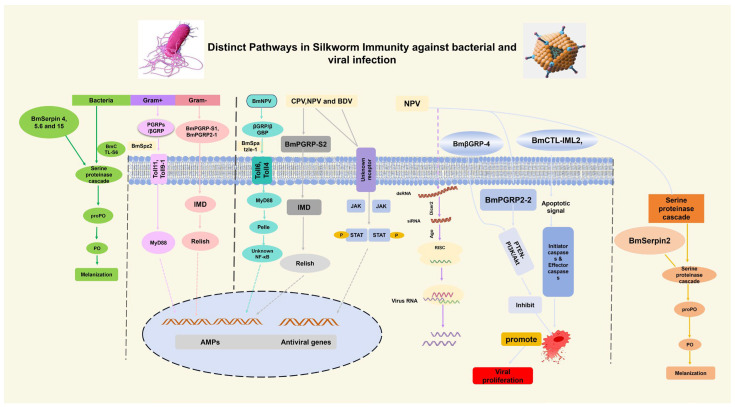
Distinct bacterial and viral-induced immune pathways, focusing on the extracellular versus the intracellular sensing, immune signaling pathways (TOLL and Imd, RNAi, JAK/STAT, Apoptosis, melanization) and effector molecules, such as AMPs.

## Data Availability

No new data were created or analyzed in this study. Data sharing is not applicable to this article.

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
