# Peer review of "Host Immunity Mechanisms Against Bacterial and Viral Infections in Bombyx mori"

_insects, 2025, doi:10.3390/insects16111167_

Round 1

Reviewer 1 Report

Comments and Suggestions for Authors

Reviewer Comments to Author

Sadaf Ayaz and colleagues report a comprehensive and well-organized review on “ Host Immunity Mechanisms against Bacterial and Viral Infections in Bombyx mori”  that offers valuable insights into the immune mechanisms of Bombyx mori. The depth of research and clear synthesis of various immune pathways make it an excellent contribution to the field. The figures are informative and enhance the overall clarity. I have, however, some minor suggestions aimed at improving the manuscript's readability and coherence before acceptance.

  1. The paper uses both "hemolymph" and "haemolymph." It would be better to consistently use "hemolymph" throughout the manuscript for uniformity.
  2. The paper references Figure 1, but there is no citation of it in the text. It would be helpful to ensure that Figure 1 is properly cited.
  3. L296-297, revise the statement.
  4. There seems to be an inconsistency between the abstract and the introduction. The abstract primarily emphasizes the role of CRISPR-Cas gene editing in enhancing disease resistance and silk quality, whereas the introduction focuses mainly on the innate immune system of Bombyx mori and how it handles bacterial and viral infections. This creates a slight disconnect between the two sections, as the introduction doesn't align with the abstract's focus on gene editing
  5. It is recommended to revise the entire paper to ensure that all gene names are italicized throughout the manuscript.
  6. Same as comment No. 2, Figures 2 and 3, there is no citation of them in the text. It is recommended to revise all figures to be cited in the paper.
  7. In line 164, the abbreviation 'PPO' is used without first defining it
  8. Line 167-169, it's hard to read, please rewrite it.
  9. Line 225, a full stop is missing
  10. There is a missing space between the word and the citation.
  11. Line 368-371. seems a bit dense. It would help to simplify the structure and revise the statement
  12. Revise the grammar of the whole review. There seems a grammatical mistake

13. According to Line 68-70, how can silkworm immunity be optimized to increase silk production in the face of infections?

  1. Is there any influence of viral and bacterial pathogens on the gut microbiota of silkworm?
  2. Line 396-397. It would be better to simplify the structure,

Author Response

Reviewer comment

We express our sincere gratitude to all reviewers and the editor for their precious time, perceptive remarks, and constructive recommendations. Their comments have significantly enhanced the quality and clarity of our writing. We have meticulously considered all feedback and amended the content accordingly. We sincerely value their contributions to improving the scientific reliability and presentation of our work.

Reviewer #1

Q1. The paper uses both "hemolymph" and "haemolymph." It would be better to consistently use "hemolymph" throughout the manuscript for uniformity.

Answer: Thank you for pointing this out. We have revised the manuscript to consistently use the term "hemolymph" throughout, ensuring uniformity in terminology(line 131 in the revised manuscript)

Q2. The paper references Figure 1, but there is no citation of it in the text. It would be helpful to ensure that Figure 1 is properly cited.

Answer: Thank you for bringing this to our attention. We have now properly cited (Figure 1, line 113,114 in revised manuscript ) in the relevant sections of the text to ensure it is referenced appropriately

Q3. L296-297, revise the statement.

Answer: We appreciate your suggestion. The statement on lines 296-297 has been revised for clarity and conciseness( line 349-352 in revised manuscript) .We believe the revised version enhances readability and better conveys the intended message.

Q4. There seems to be an inconsistency between the abstract and the introduction. The abstract primarily emphasizes the role of CRISPR-Cas gene editing in enhancing disease resistance and silk quality, whereas the introduction focuses mainly on the innate immune system of Bombyx mori and how it handles bacterial and viral infections. This creates a slight disconnect between the two sections, as the introduction doesn't align with the abstract's focus on gene editing

Answer: Thank you for your insightful observation. We have revised the introductionand added information about CRISPR-Cas to better align with the abstract, ensuring that both sections present a cohesive narrative ( line 92 to 101 in revised manuscript)

Q5. It is recommended to revise the entire paper to ensure that all gene names are italicized throughout the manuscript.

Answer: Thank you for the reminder. We have carefully reviewed the manuscript and ensured that all necessary gene names are italicized in accordance with standard scientific conventions.

Q6. Same as comment No. 2, Figures 2 and 3, there is no citation of them in the text. It is recommended to revise all figures to be cited in the paper.

Answer: We appreciate your feedback. (Figures 2 at line 158 and Figure 3 at line 382 in revised manuscript) have now been cited in the relevant sections of the text, ensuring that all figures are properly referenced in the manuscript.

Q7. In line 164, the abbreviation 'PPO' is used without first defining it

Answer: Thank you for your comments. I believe there may have been a misunderstanding. Upon reviewing the document, I did not find “PPO” in line 164 you mentioned. It seems there may have been an oversight. However, I have made sure to correct the usage of “PPO” where it was first introduced, (line 193 in revised manuscript).

Q8. Line 167-169, it's hard to read, please rewrite it

Answer: Thank you for highlighting this issue. We have rewritten lines 167-169 ( line 173-178 in revised manuscript) to improve readability and simplify the structure, making it clearer and easier to follow.

Q9. Line 225, a full stop is missing

Answer: Thank you for noticing this small detail. We have added the full stop at the end of line 225 ( line 277 in revised manuscript).

Q10. There is a missing space between the word and the citation.

Answer: Thank you for pointing out the formatting issue. We have corrected the spacing problem between the words and citations throughout the manuscript.

 Q11. Line 368-371. seems a bit dense. It would help to simplify the structure and revise the statement

Answer: We appreciate your suggestion to simplify this section. Lines 368-371 have been revised for better clarity and flow. The sentence structure has been simplified to improve readability and ensure the message is conveyed more effectively.( line 435-441 in revised manuscript).

Q12. Revise the grammar of the whole review. There seems a grammatical mistake.

Answer: Thank you for your valuable input. We have thoroughly reviewed the manuscript for grammatical errors and have made the necessary corrections to improve clarity and coherence

Q13. According to Line 68-70, how can silkworm immunity be optimized to increase silk production in the face of infections?

Answer: Thank you for your useful comments; Actually, silkworm immunity can be optimized to increase silk production by enhancing disease resistance through genetic modifications, such as CRISPR/Cas9, to boost immune pathways like RNA interference (RNAi) and antimicrobial peptide (AMP) production. Strengthening the silkworm's ability to resist bacterial and viral infections would reduce mortality and improve overall health, leading to higher silk yield and quality.

Q14. Is there any influence of viral and bacterial pathogens on the gut microbiota of silkworm?

Answer: Yes, both viral and bacterial pathogens can influence the gut microbiota of silkworms. Bacterial infections can disrupt the balance of beneficial gut bacteria, either by direct competition for resources or by altering the microbiota's composition. Similarly, viral infections may affect gut microbial diversity by modulating the immune response, potentially creating an environment that favors certain microbial species. These changes in the gut microbiota can impact the silkworm's overall health and immune function

Q15. Line 396-397. It would be better to simplify the structure.

Answer: Thank you for your suggestion. We have revised lines 396-397 and simplified the sentence structure, making the content clearer and more digestible for readers ( line 467-471 in revised manuscript).

We sincerely thank you for your constructive feedback. It has been invaluable in improving the manuscript. If you have any further suggestions or comments, please don't hesitate to let us know.

Reviewer 2 Report

Comments and Suggestions for Authors

Numerous immunity-related proteins in the silkworm have been characterized, primarily through the seminal work of Tanaka et al. (2008). To help readers quickly grasp the historical progression of silkworm immunity research, please compile a concise table listing these key proteins, their specific functions, and corresponding references.

Additionally, the manuscript suffers from abrupt transitions between paragraphs, creating noticeable gaps in logical flow. Please revise the text to ensure smooth connectivity throughout, with particular attention to seamlessly integrating the background section on B. mori immunity—currently it appears disjointed and poorly incorporated. Strengthen paragraph linkages using transitional phrases and ensure each section builds coherently on the previous one.

Fig 1 should be redrawn to illustrate the workflow of the silkworm immune response, with a primary focus on immune mechanisms. Include gut immune responses as a key component. Enlarge the silkworm illustration and integrate major immune pathways (e.g., Toll, IMD, JAK/STAT) within the body. Highlight physical barriers (cuticle, peritrophic membrane) surrounding the silkworm externally to emphasize their role in blocking most pathogens prior to immune activation. Replace GRP with βGRP, and 1,3-Glucan Recognition Proteins with β-1,3-Glucan Recognition Proteins.

In section 3.1, please provide background information for each type of PRR and describe the family members in detail, indicating which ones have been studied and which remain unexplored. For example, specify how many PGRPs, βGRPs, and LBPs are present in the silkworm, and summarize the research status of each. I also noticed that some PRRs, such as BmPGRP-S5, were not included in the manuscript; please incorporate these to make the review more comprehensive.

For Figure 2, please include the JNK and JAK/STAT pathways, and highlight the downstream effectors. For instance, indicate which AMPs are regulated by the Toll or IMD pathways. Additionally, please consider the role of Toll9, which also functions as a pattern recognition receptor (Zhang et al., 2021).

Short session 2 and move the key points into session 3. Some of them are repeated in session 3.

Some elements of Figures 2, 3, and 4 appear to be repetitive; please revise them to avoid redundancy.

Line 103: “However, the fat body and haemocytes are the primary tissues involved in insect innate immunity [17].” Since most pathogens enter the silkworm through feeding, please also note that the gut plays a key role in defending against these pathogens (https://doi.org/10.1016/j.dci.2020.103720).

Line 132: Please cite relevant references to support this section.

Author Response

Reviewer comment

We express our sincere gratitude to reviewer and the editor for their precious time, perceptive remarks, and constructive recommendations. Their comments have significantly enhanced the quality and clarity of our writing. We have meticulously considered all feedback and amended the content accordingly. We sincerely value their contributions to improving the scientific reliability and presentation of our work.

Reviewer #2

Q1. Numerous immunity-related proteins in the silkworm have been characterized, primarily through the seminal work of Tanaka et al. (2008). To help readers quickly grasp the historical progression of silkworm immunity research, please compile a concise table listing these key proteins, their specific functions, and corresponding references.

Answer: Your suggestion is very legitimate. We have compiled a concise table Section 3.1., Table 1, page 7 & 8 in revised manuscript) containing some of the most important immunity-related proteins in the Bombyx mori, their respective functions and reference in particular, the seminal work by Tanaka et al. (2008). This table now gives the reader a brief understanding of the chronological development of the silkworm immunity research and the description of these proteins.

Q2. Additionally, the manuscript suffers from abrupt transitions between paragraphs, creating noticeable gaps in logical flow. Please revise the text to ensure smooth connectivity throughout, with particular attention to seamlessly integrating the background section on B. mori immunity—currently it appears disjointed and poorly incorporated. Strengthen paragraph linkages using transitional phrases and ensure each section builds coherently on the previous one.

Answer: I appreciate your mentioning this. The manuscript has been revised to enhance the coherence and connection between paragraphs. We also included transitional phrases in order to reinforce the connection between paragraphs and make each part to be followed by the other. Also, the background section of the Bombyx mori immunity has been smoothly incorporated into the other parts of the manuscript so as to make the narrative more realistic background section of the insect immunity (line 110- 149 in revised manuscript)

Q3. Fig 1 should be redrawn to illustrate the workflow of the silkworm immune response, with a primary focus on immune mechanisms. Include gut immune responses as a key component. Enlarge the silkworm illustration and integrate major immune pathways (e.g., Toll, IMD, JAK/STAT) within the body. Highlight physical barriers (cuticle, peritrophic membrane) surrounding the silkworm externally to emphasize their role in blocking most pathogens prior to immune activation. Replace GRP with βGRP, and 1,3-Glucan Recognition Proteins with β-1,3-Glucan Recognition Proteins.

Answer: Thanks to your suggestions that were so helpful on Figure 1. We have re-drawn the figure to more effectively indicate the workflow of silkworm immune response, but with a more focus on immune mechanisms. The silkworm image has been magnified and the gut immune responses as an important element are added(Figure 1, page 4 in revised manuscript). We have also identified the physical barriers like the cuticle and peritrophic membrane to bring into focus how they can prevent entry of pathogens prior to activation of immune.

Q4. In section 3.1, please provide background information for each type of PRR and describe the family members in detail, indicating which ones have been studied and which remain unexplored. For example, specify how many PGRPs, βGRPs, and LBPs are present in the silkworm, and summarize the research status of each. I also noticed that some PRRs, such as BmPGRP-S5, were not included in the manuscript; please incorporate these to make the review more comprehensive.

Answer: Thank you very much, your feedback was detailed. We have in turn used this as the opportunity to extend Section 3.1. to provide detailed background information about each of the types of pattern recognition receptor (PRR) in Bombyx mori. We have determined the number of PGRPs, βGRPs and LBPs that are available in the silkworm and we have given a summary of the research status of each. Additionally, we have included the previously overlooked BmPGRP-S5, as you suggested, to make the review more comprehensive. These additions can be found in section 3.1. , line 198- 267, page 5 & 6 in revised manuscript).

Q5. For Figure 2, please include the JNK and JAK/STAT pathways, and highlight the downstream effectors. For instance, indicate which AMPs are regulated by the Toll or IMD pathways. Additionally, please consider the role of Toll9, which also functions as a pattern recognition receptor (Zhang et al., 2021).

Answer: Thanks a lot on this helpful suggestion. We have also revised Figure 2 to incorporate  JAK/STAT pathways, and their downstream effectors. We have also pointed out to the antimicrobial peptide (AMP) regulated by the Toll and IMD routes. Moreover, we have also mentioned the role of Toll9 as a pattern recognition receptor with a corresponding study done by Zhang et al. (2021). (see Figure 2, page 7 in revised manuscript )

Q6. Short session 2 and move the key points into session 3. Some of them are repeated in session 3.

Answer: Thanks to the mentioning of the duplication between section 2 and 3. We have cut down on Section 2 and transferred the main details to Section 3 to not repeat and make the information presented in a more concise way.

Q7. Some elements of Figures 2, 3, and 4 appear to be repetitive; please revise them to avoid redundancy.

Answer: Your comment about the appearance of repetition in Figures 2, 3, and 4 is very insightful. Attention to detail is valued at our place. We would however like to explain that, Figure 2 represents the overall outline of the immune response pathways that are triggered in the case of being infected by bacteria, where the bacteria are known to interact with the pattern recognition receptors (PRRs) and triggering the Toll and IMD pathways, which are followed by the ensuing immune signaling cascades. Figure 3 demonstrates the viral infection-induced specific immune response pathways, in which the (viral) PAMPs (e.g., viral RNA or protein) activate the same PRRs and immune signal cascades, such as Toll, IMD, and JAK/STAT etc, resulting in the generation of anti-viral responses such as antimicrobial peptides (AMPs).Both bacterial and viral immune response is more integrated as in figure 4, the two pathways overlap and interrelate at different points of immune activation. This fact highlights the common immune machinery used in pathogen recognition thus the similarities to Figures 2 and 3, although they represent different immune responses depending on the type of the infecting pathogen. Thus, the "repetition" you have observed is not redundant but instead highlights the shared immune components that are crucial for both bacterial and viral defense in the silkworm. These shared pathways underscore the centrality of Toll, IMD, and JAK/STAT signaling in regulating immune responses across different infection types.

Q8. Line 103: “However, the fat body and haemocytes are the primary tissues involved in insect innate immunity [17].” Since most pathogens enter the silkworm through feeding, please also note that the gut plays a key role in defending against these pathogens (https://doi.org/10.1016/j.dci.2020.103720).

Answer: This is a valuable suggestion that can be appreciated. We have revised the text of line 103 ( line 115-117, page 3 in revised manuscript ) to highlight the importance of the gut in preventing pathogen invasion because the majority of the pathogens ingress Bombyx mori during feeding. It is the reference (https://doi.org/10.1016/j.dci.2020.103720) that keeps being relevant.

This addition has been supported by citing.

Q9. Line 132: Please cite relevant references to support this section.

Answer: Your recommendation is appreciated. We have included the corresponding references to substantiate the text on line 132. These sources make the manuscript more scientific and add more background to the arguments ( line 146, page 4 in revised manuscript).

We sincerely thank you for your insightful and constructive feedback. It has been invaluable in improving the manuscript. If you have any further suggestions or comments, please do not hesitate to let us know.

Round 2

Reviewer 2 Report

Comments and Suggestions for Authors

Thanks for your revision, all my questions were answered.

Please unify the reference format, see below

159. Wang, X.-y.; Ding, X.-y.; Chen, Q.-y.; Zhang, K.-x.; Zhao, C.-x.; Tang, X.-d.; Wu, Y.-c.; Li, M.-w. Bmapaf-1 is involved in the response 1094 against BmNPV infection by the mitochondrial apoptosis pathway. Insects 2020, 11, 647. 1095
160. Wang, X.; Zhao, Z.q.; Huang, X.m.; Ding, X.y.; Zhao, C.x.; Li, M.w.; Wu, Y.c.; Liu, Q.n.; Wang, X.y. Bmcas‐1 plays an important role 1096 in response against BmNPV infection in vitro. Archives of Insect Biochemistry and Physiology 2021, 107, e21793. 1097
161. Wang, X.-Y.; Wu, K.-H.; Pang, H.-L.; Xu, P.-Z.; Li, M.-W.; Zhang, G.-Z. Study on the role of cytc in response to BmNPV infection in 1098 silkworm, Bombyx mori (Lepidoptera). International journal of molecular sciences 2019, 20, 4325.